# Neuroprotective Effects of an Aqueous Extract of *Forsythia viridissima* and Its Major Constituents on Oxaliplatin-Induced Peripheral Neuropathy

**DOI:** 10.3390/molecules24061177

**Published:** 2019-03-25

**Authors:** Jin-Mu Yi, Sarah Shin, No Soo Kim, Ok-Sun Bang

**Affiliations:** Clinical Medicine Division, Korea Institute of Oriental Medicine, Daejeon 34054, Korea; jmyi@kiom.re.kr (J.-M.Y.); s.sarah@kiom.re.kr (S.S.); nosookim@kiom.re.kr (N.S.K.)

**Keywords:** *Forsythia viridissima*, arctigenin, matairesinol, oxaliplatin, oxaliplatin-induced peripheral neuropathy, neuroprotection, von Frey test

## Abstract

The dried fruits of *Forsythia viridissima* have been prescribed to relive fever, pain, vomiting, and nausea in traditional medicine. Oxaliplatin (LOHP) is used to treat advanced colorectal cancer; however, it frequently induces peripheral neuropathies. This study was done to evaluate the neuroprotective effects of an aqueous extract of *Forsythia viridissima* fruits (EFVF) and its major constituents. Chemical constituents from EFVF were characterized and quantified with the UHPLC-diode array detector method, and three major constituents were identified as arctiin, matairesinol, and arctigenin. The in vitro cytotoxicity was measured by the Ez-cytox viability assay, and the in vivo neuroprotection activity was evaluated by a von Frey test in two rodent animal models that were administered LOHP. EFVF significantly alleviated the LOHP-induced mechanical hypersensitivity in the induction model. EFVF also prevented the induction of mechanical hyperalgesia by LOHP in the pre- and co-treatment of LOHP and EFVF. Consistently, EFVF exerted protective effects against LOHP-induced neurotoxicity as well as inhibited neurite outgrowths in PC12 and dorsal root ganglion cells. Among the major components of EFVF, arctigenin and matairesinol exerted protective effects against LOHP-induced neurotoxicity. Therefore, EFVF may be useful for relieving or preventing LOHP-induced peripheral neuropathy in cancer patients undergoing chemotherapy with LOHP.

## 1. Introduction

Chemotherapy-induced peripheral neuropathy (CIPN) is a common side effect that is caused by exposure to neurotoxic chemotherapeutic agents such as platinum analogs, taxanes, vinca alkaloids, and bortezomib [1,2]. Clinical symptoms of CIPN can range from a transient numbness and tingling to a severe chronic impaired sensation and burning in the hands or feet with a “stocking-glove” distribution [3,4]. Oxaliplatin (LOHP) is a platinum-based chemotherapy drug that has been used to treat advanced colorectal cancer. However, LOHP induces a side effect which is acute or chronic peripheral neuropathy [5]. A transient and acute neurotoxicity appears during or shortly after chemotherapy and it may soon recover. Chronic oxaliplatin-induced peripheral neuropathy (OIPN) is a dose limiting and cumulative neurotoxicity, which is characterized by distal paresthesia and numbness, and takes months or years to resolve and or even persists throughout life with a substantial effect on quality of life and functional status [6,7,8]. 

Understanding the mechanisms underlying acute and chronic OIPN will be a key component of future advances in the prevention and treatment of OIPN; however, these mechanisms have not been fully elucidated, and no preventive or therapeutic agents for OIPN are available yet. Therefore, only symptomatic management is being used, which includes antidepressants and anticonvulsant drugs including tricyclic antidepressant, carbamazepine, gabapentin (GBP), and venlafaxine, but they have been mostly ineffective, or their pharmaceutical efficacies were not validated in clinical trials. In addition, a recent systematic review reported that there was insufficient evidence to confirm the efficacy of central nervous system drugs for CIPN [9,10,11,12]. To date, there are no effective strategies to prevent CIPN, and there is only limited evidence on effective drugs for treating established CIPN. Only duloxetine has been shown to provide clinically meaningful neuropathy relief in patients with chronic CIPN [13]. Thus novel effective therapeutic strategies are urgently required for the prevention and treatment of CIPN. 

In the Korean Pharmacopoeia Forsythiae Fructus (FF) is the dried fruits of *Forsythia suspensa* Vahl. and *Forsythia viridissima* Lindley, both of which are ornamental deciduous shrubs belonging to the Oleaceae family [14]. FF has been widely known as a heat-clearing and detoxifying medication in Traditional China and Korea Medicine [15,16,17]. Various pharmacological studies have revealed that FF possesses antioxidant [18], antiviral [19], anti-inflammatory [20,21], antitumor [22], and anti-allergy [23] effects. To reveal their pharmacological effects, the phytochemical constituents such as lignans, flavonoids, and phenylethanoid glycosides were isolated from FF, and their related biological effects were elucidated [16,19,20,24,25,26]. However, up to now, most of the studies on the chemical compounds as well as the pharmacological evaluation and quantitative analyses have been done on *F. suspensa*. The pharmacological values and chemical compositions of *F. viridissima* have not been elucidated fully. 

Because the agents currently used to relieve peripheral neuropathy have little effect, and their side effects are serious, a novel drug needs to be developed that alleviates OIPN with minimum adverse side effects. We have tried to find effective materials to relieve LOHP-mediated neurotoxicity using a library of traditionally used medicinal herb extracts. This study shows that the aqueous extract of *F. viridissima* (EFVF) and its major constituents have neuroprotective potential against OIPN in both in vitro and in vivo neuropathic models.

## 2. Results

### 2.1. Ultra-High Performance Liquid Chromatography Analysis of EFVF

EFVF was obtained by water extraction of *F. viridissima* fruits (16.8% yield). To analyze the chemical constituents of EFVF, we optimized the ultra-high performance liquid chromatography -diode array detector (UHPLC-DAD) analytical conditions including the mobile phase, column, sample diluent, and thermostat temperature. A mobile phase system using 0.1% (*v/v*) formic acid (mobile phase A) and acetonitrile (mobile phase B) improved the peak resolution and tailing (Figure 1A, upper panel). To determine the major constituents of EFVF, its UHPLC chromatogram was scanned and compared to the peaks of authentic standard chemicals (STDs). Since several lignans from the methanol extract of *F. viridissima* were isolated and their biological activities have been studied [27,28], we first compared the DAD spectra and retention times (t_R_) data between EFVF peaks and standard chemicals of lignan family, including arctiin, arctigenin, matairesinol, phillygenin, phillyrin, pinoresinol, (+) pinoresinol-β-D-glucopyranoside, caffeoyl pinoresinol and lariciresinol. Among them, the three major peaks of EFVF showed typical DAD spectra similar to those of lignans such as arctiin (AT), matairesinol (MTR), and arctigenin (ATG). The t_R_ of AT, MTR, and ATG in the overlaid chromatogram of STDs (Figure 1A, lower panel) were 15.91, 18.90, and 21.98 min, respectively, and they matched those of the three major constituent peaks of EFVF. In addition, the DAD spectra in 210–450 nm spectral scanning also showed a perfect matching between the three constituents of EFVF and their STDs (Appendix A). Therefore, the three major peaks observed in EFVF chromatogram were identified as AT, MTR, and ATG, respectively, and their structures are shown in Figure 1B.

To quantify the three major constituents in EFVF, we obtained the calibration curve of the three STDs with a linearity r^2^ = 0.9998 in a range of 5 to 200 μg/mL. The other calculated quantitation parameters are presented in Table 1. The contents of the AT, MTR, and ATG in the EFVF were measured as 22.84 ± 0.05, 59.31 ± 0.09, and 23.23 ± 0.05 mg/g, respectively. 

### 2.2. EFVF Attenuated LOHP-Induced Peripheral Neuropathy in Animal Models

To evaluate the effect of EFVF on peripheral neuropathy, we first developed an animal model of LOHP-induced neurotoxicity. Sprague-Dawley rats received 5 mg/kg of LOHP twice per week for 3 weeks (total 30 mg/kg) to induce peripheral neuropathy. Among them, the rats showing mechanical hypersensitivity in the von Frey test were divided into three groups and then, each group received EFVF (100 mg/kg), GBP (positive control, 50 mg/kg) or 0.5% (*w/v*) sodium-carboxymethyl cellulose solution (CMC, negative control). During the entire experimental period, no clinical signs related to toxicity such as extensive loss of body weight were observed in all the animal groups. However, body weight gain was decreased in the LOHP-treated rats compared to those in the vehicle group, and body weight gain did not change by the co-treatment with EFVF or GBP (Figure 2A). Next, to examine the hyperalgesic effect of LOHP, mechanical allodynia was evaluated by measuring the nociceptive withdrawal threshold using the electronic von Frey tester. As depicted in Figure 2B, the withdrawal threshold of rats was dramatically decreased to 14.5 ± 0.4 g by the LOHP treatment, which persisted over 40 days. However, the untreated vehicle group consistently maintained the levels of withdrawal threshold at 22.5 ± 1.4 g. When the rats were administered with EFVF 3 weeks after the LOHP injection, the decreased withdrawal threshold by the LOHP was increased to 18.8 ± 4.7 g one week after the treatment and was maintained at about 20.4 ± 1.2 g to the end of the study. GBP, a blocker of the calcium channel subunit a2d1 has been known to attenuate LOHP-induced mechanical hyperalgesia in in vivo models [29]. As expected, GBP treatment significantly increased the withdrawal threshold up to 24.4 ± 3.2 g during treatment and then, dramatically decreased again to the basal level after termination of the treatment (Figure 2B). In addition, there was a significant interaction between treatment group and time (two-way repeated measures ANOVA, *p* < 0.001). Because the loss of intraepidermal nerve fibers (IENFs) in the peripheral nerve is known to be related with a pathological finding in cancer patients receiving chemotherapy [30], we investigated whether LOHP can induce the denervation of peripheral nerves in distal foot skin by immunostaining nociceptive IENFs with anti-protein gene product 9.5 (PGP9.5) antibody (Figure 2C). In untreated vehicle rats, the IENFs were perpendicular to the epidermis and regularly penetrated through the epidermal-dermal junctions. The density of the IENFs in the epidermis was quantified by digital analysis of the immunostained sections. Animals receiving LOHP showed a remarkable loss of IENFs by 57.5 ± 7.5% when compared to vehicle treated rats. However, such loss of IENFs by LOHP was significantly rescued by the treatment with EFVF (by 34.2 ± 11.6%) but not with GBP.

To investigate whether EFVF can inhibit the induction of peripheral neuropathy by LOHP, we also established a prevention mouse model as follows: Mice were daily administered in advance with EFVF (50 mg/kg) or 0.5% CMC for a week and then, received EFVF or 0.5% CMC in combination with LOHP (10 mg/kg, once a week, three times). Body weights were slightly decreased in the LOHP treated groups (Figure 3A). The pain sensitivity was measured by the von Frey test as shown in Figure 3B. The incidence rates of the mechanical hypersensitivity in each treatment group were calculated by counting the number of mouse that withdrew the left or right hind paw in response to mechanical stimuli. LOHP, 2–3 weeks after treatment, dramatically increased the incidence rates of the hypersensitivity to mechanical stimuli by von Frey filaments with 0.16 and 0.4 g bending forces up to 50.0 and 75.0–79.2%, respectively, while the untreated vehicle mice showed response rates below 20.8 and 37.5%, respectively. Interestingly, co-treatment of EFVF (50 mg/kg) reduced the mechanical hypersensitivity induced by LOHP throughout the experiment: the incidence rates were decreased again to basal level compared to the vehicle mouse (Table 2). Taken together, these data indicate that consecutive treatments of LOHP produced hypersensitivity to mechanical stimulation and loss of IENFs in peripheral nerves, which can be recovered by EFVF treatment.

### 2.3. EFVF Relieved LOHP-Induced Cytotoxicity in Neuronal Differentiated PC12 

To investigate the effect of EFVF on PC12 cell viability, cells were cultured in a growth medium without nerve growth factor (NGF) and then, treated with various concentrations of EFVF for 48 h. The cell viability was measured by with Ez-Cytox viability assay. As shown in Figure 4A, EFVF treatment did not change the cell viability of PC12 cells at a range from 0 to 100 μg/mL. Because the cytotoxic effect of LOHP in PC12 cells has been reported [31,32], we investigated the effect of EFVF on the LOHP-induced cytotoxicity. PC12 cells were cultured in a differentiation medium containing NGF, and treated with a combination of LOHP (200 nM) and the indicated concentration of EFVF for 48 h. LOHP critically reduced cell viability to 68.8 ± 1.5% in PC12 cells, which was not affected by co-treatment of EFVF (Figure 4B). To examine the effect of EFVF on the neurotoxicity of LOHP in differentiated PC12 cells (neural PC12), the cells were pre-cultured in a differentiation medium containing NGF (100 ng/mL) for 96 h to induce neuronal differentiation and then, treated with LOHP (200 nM) alone or co-treated with the indicated concentrations of EFVF for a further 48 h. LOHP dramatically reduced the viability of neural PC12 cells to 43.6 ± 11.6% compared to the untreated vehicle. However, EFVF attenuated the LOHP-induced neurotoxicity in a dose-dependent manner (Figure 4C). Amifostine (0.5 mM) was used as a positive control.

To investigate whether EFVF could affect the LOHP antitumor activity, human colorectal (HCT116, KM12SM and COLO205) and breast (MDA-MB-468 and MCF7) cancer cells were exposed to various concentrations LOHP (0–100 μM) in the presence of EFVF (50 and 100 μg/mL) for 48 h, and the cell proliferation was measured. As shown in Table 3, the IC_50_ values of LOHP were not significantly changed by EFVF meaning that EFVF could exert its protective potential against LOHP-induced neurotoxicity without affecting the LOHP antitumor activity. 

### 2.4. EFVF Attenuated LOHP-Induced Neurotoxicity in PC12 Cells.

Next, we investigated the effect of LOHP and EFVF on neurite outgrowth in neural PC12 cells. PC12 cells were cultured in NGF-containing differentiation medium and then treated with LOHP alone or in combination with the EFVF (25, 50 and 100 μg/mL). PC12 cells cultured with NGF, sprouted out extensive neurites from the cell bodies and developed synaptic-like vesicles shown in Figure 5C. The number of PC12 cells bearing neurites that were longer than the cell body were counted and compared to that of the untreated vehicle. 

LOHP induced a strong neurotoxicity, which was determined by the decreases in both the number of neurite-bearing cells and the total length of the neurites (45.8 ± 5.4% and 40.4 ± 6.1%, respectively), compared with vehicle. The EFVF co-treatment reduced the LOHP-induced neurotoxicity in the PC12 cells in a dose-dependent manner, and the highest dose of EFVF (100 μg/mL) alleviated the LOHP-induced neurotoxicity above 95% in based on both the number of cells bearing neurites and the total neurite lengths (Figure 5A,B).

### 2.5. Effect of the Major Constituents of EFVF on LOHP-Induced Neurotoxicity

As described earlier, the three major constituent peaks of the EFVF chromatogram were identified as MTR, AT and ATG. To investigate the effects of the three major constituents on PC12 cell viability, cells were cultured in a growth medium without NGF and then, treated with the indicated concentrations of MTR, AT, or ATG (0–50 μM) for 48 h. In undifferentiated PC12 cells, the three constituent did not affect the cell viability at a concentration between 0–10 μM. However, the viability of cells treated with 50 μM MTR or AT was decreased to by 21.1 ± 7.6% and 28.8 ± 6.3%, respectively, and 50 μM ATG treatment remarkably decreased the cell viability by 89.9 ± 4.8% (Figure 6A). Therefore, we examined the effect of the three major constituents at 0–10 μM on the LOHP-induced neurotoxicity in the neural PC12 cells in the presence of NGF (100 ng/mL). When the cells were co-treated with AT and LOHP (200 nM), the cell viability was significantly increased at concentrations of 2–10 μM. Interestingly, co-treatment of ATG with LOHP remarkably attenuated the LOHP-induced neurotoxicity in a dose-dependent manner to a similar level of the EFVF treatment. However, MTR did not affect the LOHP-induced neurotoxicity (Figure 6B). 

To investigate the effect of the major constituents on neurite outgrowth, PC12 cells were cultured in NGF-containing differentiation medium and then, treated with LOHP alone or in combination with the indicated concentrations of MTR, AT, or ATG (0.4, 2, and 10 μM). The number of neural PC12 cells bearing neurites and the total length of the neurites were counted as described earlier. As shown in Figure 7, co-treatment with MTR or ATG remarkably attenuated the LOHP-induced neurotoxicity above 80% based on both the number of cells bearing neurites and the total length of neurites. Surprisingly, ATG at 10 μM recovered cells to almost 100% from the LOHP-induced neurotoxicity (Figure 7A,B). The results indicate that among the three constituents, ATG could exert a strong protective effect against LOHP-induced neurotoxicity in neural PC12 cells.

### 2.6. Protective Effect of EFVF on LOHP-Induced Neurotoxicity in DRG Cells

To investigate the effect of EFVF on neurite outgrowth and the viability of primary neuron cells, DRG cells were isolated from C57BL/6 mice. DRG cells (2 × 10^5^), suspended in NGF containing Neurobasal plus medium supplemented with B-27 (Neurobasal/B-27 medium), were plated into Millicell inserts and treated with LOHP (30 μM) alone or in combination with EFVF (25 or 50 μg/mL). After 7 days of culturing, the neurites sprouting out from the DRG cells were stained with Neurite Stain Solution (Figure 8B). The stained neurites were extracted, and the neurite outgrowth was quantitated using a multiplate reader. LOHP decreased the neurite outgrowth of the DRG cells to 23.2 ± 13.0% compared to saline-treated cells, which was dose-dependently restored by the co-treatment of EFVF (Figure 8A). The data indicate that EFVF alleviates LOHP-induced neurotoxicity in primary DRG cells. In addition, mouse DRG cells were cultured with NGF and treated with LOHP and EFVF for 48 h, and cell viability was measured with the Ez-Cytox viability assay. As shown in Figure 9A. LOHP (30 μM) significantly decreased the cell viability of the primary DRG cells to 43.1 ± 5.6%, which was restored by the co-treatment with EFVF (25 and 50 μg/mL) to 61.8 ± 14.3% and 77.6 ± 8.8%, respectively.

### 2.7. Protective Function of EFVF on LOHP-Induced Apoptosis 

To investigate whether the LOHP-induced neurotoxicity in DRG cells is caused by apoptosis, the DRG cells were cultured in NGF containing Neurobasal/B-27 medium for 7 days and treated with LOHP alone or in combination with EFVF for 3 days. Apoptotic cell staining was performed using a FITC-Annexin V apoptosis kit. As seen in Figure 9B, the saline-treated cells were healthy (double negative), while the LOHP treatment increased the number of apoptotic cells (double positive). However, co-treatment of EFVF with LOHP reduced the number of double positive apoptotic cells indicating that LOHP induces apoptotic cell death in neuronal cells which can be rescued by the co-treatment of EFVF. 

Mitochondrial function, a key indicator of cell health, can be assessed by monitoring the changes in the mitochondrial membrane potential (MMP). It has been reported that a perturbation in mitochondria function reduces the MMP and the decrease in the MMP may also be linked to apoptosis [33,34]. To understand the molecular mechanism of the EFVF effect on cell viability and neuronal differentiation, the mitochondrial function of the cells was determined by analyzing the change in MMP using the JC-1 dye. In healthy cells, JC-1 accumulates in the mitochondria as red fluorescent aggregates (emission at 590 nm). When the MMP depolarizes and cells become less healthy, the JC-1 aggregates are converted to green fluorescent monomers (emission at 535 nm) and remain in the cytoplasm. As shown in Figure 9C, the LOHP treatment reduced the JC-1 aggregates but increased the JC-1 monomers in the cytoplasm of the DRG cells, whereas most of the vehicle-treated cells produced JC-1 aggregates. Co-treatment with EFVF inhibited the LOHP-mediated JC-1 monomeric conversion to the level observed in the vehicle-treated cells. Next, we examined whether the molecular mechanism of these mitoprotective effects of EFVF against the LOHP-induced neurotoxicity in the DRG cells is related to regulating the production of reactive oxygen species (ROS). As shown in Figure 9D,E, the LOHP treatment dramatically increased the ROS production, while the EFVF treatment did not affect the increased level of ROS production induced by LOHP. Taken together, our data show that EFVF could protect neurons from apoptotic cell death caused by the mitochondrial dysfunction which is induced by neurotoxic LOHP, and its protective activity is attributed to other unknown mechanism(s) rather than ROS regulation.

## 3. Discussion

LOHP is used in treating advanced colorectal cancer, and its anticancer activity is related to the induction of neurotoxicity. More than 90% of cancer patients experience acute symptoms which resolve within a few days, but almost 50% of patients suffer from chronic OIPN for months or even years [8,35]. OIPN is recognized as a dose-limiting complication and causes significant morbidity. Although there is a remarkable understanding of the acute and chronic phases of OIPN, and a variety of pharmacological approaches have been studied to prevent or treat OIPN, there are still no pharmaceutical agents available with an acceptable efficacy to manage OIPN and other CIPNs. We, therefore, tried to develop a phytoceutical drug targeting OIPN from medicinal plants that have traditionally been used. The present study is one of the preclinical studies showing the protective effect of the traditionally used medicinal herb, *F. viridissima* fruits on LOHP-induced neurotoxicity. In this study, we found that the EFVF attenuate LOHP-induced mechanical hyperalgesia in both prevention and treatment animal models. In addition, EFVF exerted a dose-dependent protective effects against LOHP-induced neurotoxicity in neural PC12 as well as in primary DRG cells. In addition, we also demonstrated that ATG and MTR, the major constituents of EFVF also exerted a strong protective effect against LOHP-induced neurotoxicity. Although the detailed molecular mechanisms underlying the anti-OIPN potential of EFVF are unclear, we found that EFVF can protect neurons from the neurotoxicity of LOHP, in part through the recovery of apoptosis caused by mitochondrial membrane dysfunction. Our results indicate that EFVF is useful for relieving OIPN and could be a promising candidate drug for the treatment of cancer patients undergoing chemotherapy with LOHP. 

Chronic neuropathy after repeated administration of LOHP leads to dysfunction of sensory neurons [36] and the degree of OIPN is dependent on the cumulative dose, duration of administration, and dose intensity of the platinum compounds in the DRG, inducing neuronal atrophy and apoptosis [5,37,38]. To investigate the effect of EFVF on OIPN, we established chronic rat and mouse models in the absence of a tumor load for ethical reasons and practical feasibility as described in previously reported studies [39,40]. LOHP was administered at a dose of 10 mg/kg, three times to C57BL/6 mice (total 30 mg/kg) or at a dose 5 mg/kg, six times to rat (total 30 mg/kg). Using these OIPN models, we confirmed that the animals receiving LOHP could exhibit mechanical allodynia as one of the typical sensory symptoms which are assessed by applying stimuli to the hind paws of the rodents and then measuring the withdrawal responses to those stimuli using the von Frey test. The von Frey test in the OIPN rat model showed that the LOHP treatment resulted in a dose-dependent reduction in mechanical hyperalgesia and it persisted for more than 40 days, which mimics the chronic OIPN in cancer patients showing enhanced sensitivity to nociceptive mechanical stimuli. This LOHP-induced nociceptive hypersensitivity was dramatically attenuated by the treatment of EFVF. GBP, used as a positive control also attenuated the LOHP-induced nociceptive hypersensitivity however, the analgesic effects of GBP were not long lasting and declined sharply after drug withdrawal. On the other hand, the effect of EFVF persisted during the experimental period. IENF is an indicator of small fiber neuropathy and the loss of IENFs in the digitals is known to be a pathological finding in cancer patients experiencing numbness and loss of vibratory senses [30,41]. In the present study, we also observed a significant loss of IENFs in the skin of LOHP-treated rats and the loss of IENFs by LOHP was significantly recovered by the EFVF treatment. We also evaluated the protective effect of EFVF against OIPN in the prevention mice model in which mice were pre-administered with EFVF for a week, and then received LOHP (once a week, for three weeks) with or without EFVF. LOHP produced a peripheral neuropathic pain showing a higher response rate of the hind paw to the mechanical stimuli of the von Frey monofilaments of 0.16 and 0.4 g. Interestingly, daily-based oral administration of EFVF successfully blocked the induction of nociceptive mechanical hypersensitivity by LOHP. Taken together, our data indicate that EFVF not only protects against LOHP-induced peripheral effects but also has preventive effects 

It is obvious that the development of effective drugs to prevent or treat OIPN is largely dependent on understandings the mechanisms in which a disease progresses. Although the mechanisms underlying OIPN have not been fully elucidated, some putative mechanisms of OIPN have been suggested such as the release of interleukins, activation of the innate immune system and pro-inflammation, altered function of ion channels, and neuronal cell membrane remodeling [42]. Chronic OIPN has been attributed to oxidative stress, mitochondrial dysfunction, and axonal degeneration from drug accumulation in DRG neurons [43]. Because the administered LOHP accumulates and exerts a neurotoxicity in DRG where sensory neurons are located [44], DRG neurons are a primary target of LOHP-mediated peripheral neurotoxicity. In this study, we examined whether EFVF affects the neurotoxicity using PC12 and murine primary DRG cells. LOHP induced cytotoxicity in both undifferentiated and differentiated neural PC12 cells. However, EFVF treatment attenuated the LOHP-induced neurotoxicity only in differentiated neural PC12 and DRG cells. In addition, LOHP induced neurotoxicity which was determined by observing the decreases in the number of cells bearing neurites and the total length of neurites in both neural PC12 and DRG cells. EFVF treatment reduced the LOHP-induced neurotoxicity of neural PC12 cells in a dose-dependent manner, and a high dose of EFVF (100 μg/mL) attenuated LOHP-induced neurotoxicity to above 95% in both cells based on the number of cells bearing neurites and the total length of the neurites. Our data suggest that EFVF can exert its protective effect in only neuronal cells but not in other cells such as undifferentiated PC12 cells and cancer cells. We also tested whether the three major constituents of EFVF, MTR, AT, or ATG could exert a protective effect on LOHP-induced neurotoxicity. Among them, MTR and ATG remarkably attenuated the LOHP-induced neurotoxicity to above 80 % shown by the the number of cells bearing neurites and by the total length of neurites compared with vehicle. Surprisingly, ATG at 10 μM recovered almost 100% of the LOHP-induced neurotoxicity. Based on the literature, ATG and its glycoside, AT, have a low toxicity and not many side effects, and ATG has a stronger pharmacological activity than that of AT. ATG has been reported to possess a number of pharmacological properties such as antitumor, anti-inflammatory and neuroprotective effects [45,46,47,48,49]. Similarly, MTR exerts an antitumor and anti-inflammatory activity against chronic neuro- inflammation [50,51]. Until recently, however, the protective effects of these chemicals against CIPN have not been reported yet. Therefore, as far as we know, our data show for the first time that EFVF comprised of ATG and MTR has a potent protective effect against LOHP-induced neurotoxicity.

During the development of cancer supportive drugs, it is essential that the interaction between the two drugs to be co-administered should be evaluated in advance, because a supportive drug may affect the antitumor activity of chemotherapeutic agents. In the present study, we showed that co-administered EFVF did not negatively affect the antitumor activity profiles of LOHP at least in human colorectal and breast cancer cells. Therefore, it can be expected that EFVF can attenuate OIPN without affecting the anti-tumor effects of LOHP itself. However, this herb-drug interaction in terms of the anti-tumor effect of LOHP should be confirmed pre-clinically in suitable tumor-bearing animal models as well as clinically in human cancer patients.

Until now, the precise molecular mechanisms underlying the development of OIPN remains unclear and have to be elucidated to develop effective treatment or preventive drugs for OIPN. Mitochondrial dysfunction has been suggested as a significant contributor to CIPN, and the occurrence of atypical swollen and vacuolated mitochondria in peripheral nerve axons shown by electron microscopy analysis has been reported [52]. Because mitochondria are a major source of ROS and increased ROS was observed in mouse lumbar DRG after chronic LOHP treatment [53], elevated ROS production can be a consequence of mitochondrial dysfunction. It has been reported that a perturbation in mitochondria reduces the mitochondrial membrane potential (MMP), which may be linked to apoptosis [33,34]. In this study, our data show that LOHP induced the apoptosis of DRG cells, assessed by the double stain of Annexin V and PI, and EFVF treatment reversed the LOHP-induced apoptosis. When the mitochondrial function of DRG cells was determined by analyzing the change in the MMP using the JC-1 dye, LOHP treatment reduced the JC-1 aggregates but increased the JC-1 monomers in the cytoplasm, and co-treatment with EFVF reduced the LOHP-induced JC-1 monomeric conversion to the level of the vehicle-treated cells, indicating that EFVF can rescue DRG cells from the LOHP-induced mitochondrial membrane depolarization leading to apoptotic cell death. To understand the molecular mechanism of the relieving effect of EFVF on the LOHP-induced neurotoxicity, the ROS production in PC12 cells after treatment with LOHP alone or in combination with EFVF were measured. As expected, LOHP increased ROS production, while EFVF treatment did not affect the level of the ROS production. Therefore, the neuroprotective activity of EFVF in the LOHP-induced neurotoxicity is presumed to be due to other mechanisms and not the regulation of ROS generation. In conclusion, although the precise mechanism of EFVF in OIPN is unclear, in this study we demonstrated that EFVF could alleviate LOHP-induced peripheral neuropathy and suggest that it could be a starting candidate to develop an adjuvant drug for colorectal cancer patients undergoing chemotherapy in the future. 

## 4. Materials and Methods 

### 4.1. Chemicals 

STDs, such as AT, ATG, and MTR with a purity > 98%, were supplied from ChemFaces (Wuhan, Hubei, China). UHPLC-grade formic acid and solvents including water, acetonitrile, and methanol were purchased from Fisher Scientific Ltd. (Loughborough, UK). All chemicals for the other experiments, except those described elsewhere, were purchased from the Sigma-Aldrich Co (St. Louis, MO, USA). All reagents and culture wares, except those described elsewhere, were purchased from Thermo Fischer Scientific (Waltham, MA, USA).

### 4.2. Plant Materials and Extract Preparations

The dried fruits of *F. viridissima* were supplied by Kwangmyungdang Medicinal Herbs Co. (Ulsan, Republic of Korea) and identified by Dr. Goya Choi (Herbal Medicine Research Center, Korea Institute of Oriental Medicine, KIOM). A voucher specimen (KIOM-CRC#518) was stored in the Clinical Medicine Division of KIOM, Republic of Korea. The dried materials (1000 g) were extracted for 3 h, twice in a water refluxed system (10 L) and filtered through a mesh strainer and then cotton wool. The filtrates were further concentrated using a rotary evaporator (N-1200A, Rikakikai, Tokyo, Japan) and dried using a freeze dryer (FD8518, IlshinBioBase, Dongduchun, Republic of Korea). The final aqueous extracts of *F. viridissima* fruits (EFVF) were homogenized and stored in an air-tight container under desiccated condition. For the in vitro studies, EFVF was dissolved at 4 mg/mL in sterile phosphate-buffered saline (PBS) as a stock solution and stored at −80 °C. For the animal studies, EFVF was suspended in a sterile 0.5% CMC solution right before use. 

### 4.3. Chromatographic Analysis

#### 4.3.1. Chromatographic Condition 

The chromatographic analysis of EFVF was performed using a 1290 infinity UHPLC- DAD system (Agilent Technologies, Santa Clara, CA, USA) equipped with an analytical column (Luna Omega C_18_, 2.1 × 50 mm, 1.6 μm, Phenomenex, Torrance, CA, USA). The extract or STDs were analyzed using a sequential gradient mobile phase system from a 95:5 mixture to a 40:60 mixture of 0.1% (*v/v*) formic acid (mobile phase A) and acetonitrile (mobile phase B) within 40 min at a flow rate of 0.2 mL/min. The DAD signals of EFVF and STDs were detected at 280 nm, and the spectral scanning was performed at a wavelength range of 200–400 nm. The column and auto-sampler temperature were set at 40 and 4 °C, respectively. Each constituent from EFVF was identified by comparing the retention time (tR) and specific DAD spectrum pattern of each peak in the EFVF with those of the corresponding STD. The chromatographic data were processed by the Agilent OpenLAB CDS software

#### 4.3.2. Quantitative analysis 

For UHPLC analysis, the EFVF sample solution and STDs stock solutions were prepared at a concentration of 5 mg/mL and 1 mg/mL in 50% (*v/v*) methanol in water, respectively. The solutions were filtered through a 0.22 μm syringe filter (Millipore, Burlington, MA, USA) to remove impurities that interfere with UHPLC analysis. The calibration curve of each STD was obtained by the analysis of least five serial concentrations and in triplicate, and then, calculated by plotting the peak areas versus the concentration of STDs. The limit of detection (LOD) and limit of quantitation (LOQ) were calculated from the calibration curve parameters and determined with the following equations:
$$\mathrm{LOD}\;\left(\mu g/\mathrm{mL}\right)=3.3\times \frac{\mathrm{Standard}\;\mathrm{deviation}\;\left(\mathrm{SD}\right)\;\mathrm{of}\;\mathrm{the}\;\mathrm{response}}{\mathrm{Slope}\;\mathrm{of}\;\mathrm{the}\;\mathrm{calibration}\;\mathrm{curve}}$$
$$\mathrm{LOQ}\;\left(\mu g/\mathrm{mL}\right)=10\times \frac{\mathrm{SD}\;\mathrm{of}\;\mathrm{the}\;\mathrm{response}}{\mathrm{Slope}\;\mathrm{of}\;\mathrm{the}\;\mathrm{calibration}\;\mathrm{curve}}$$
The quantity of each constituent in EFVF was calculated based on the calibration curve of each STD.

### 4.4. Experimental Animals and Drug Administration 

The six-week-old male mice (C57BL/6) and five-week-old male rats (Sprague Dawley) were purchased from OrientBio, Inc (Seongnam, Korea). They were maintained in a specific-pathogen-free laboratory animal care facility with a temperature of 22 ± 2 °C, humidity of 45 ± 10% and 12 h/12 h dark/light cycle. The animals had free access to food and water and were acclimated for 1 week before the experiments. All procedures for animal care and experiments were reviewed and approved by the Institutional Animal Care and Use Committee (Protocol #17-056, 18-005, 18-036) of KIOM. 

In Korean traditional herbal medicine, 3~15 g of Forsythiae Fructus has been generally recommended as a daily dosage for heat clearing and detoxifying medication [54]. Based on the average person who weighs 60 kg, it actually taken about 8.4~41 mg/kg of EFVF (16.8% yield). Therefore, we chose about 50~100 mg/kg as a starting dose for murine studies, which is equivalent to about 8.4 mg/kg in an adult human [55].

To induce peripheral neuropathy in the rats, LOHP (5 mg/kg in 5% dextrose solution) was injected intravenously (i.v.) twice per week, for 3 weeks, and then, rats were divided into four groups. Each group (*n* = 6) received a peroral (p.o.) administration of 5% dextrose solution (vehicle control), EFVF (100 mg/kg), GBP (50 mg/kg, positive control), or 0.5% CMC solution for LOHP alone, 6 times per week, for 5 weeks.

For the prevention study, mice were divided into 3 groups (each group, *n* = 12). Group 1 (vehicle control) daily received oral administration of 0.5% CMC throughout the experiment. Group 2 and group 3 daily received oral administration of 0.5% CMC and EFVF (50 mg/kg) throughout the experiment, respectively. One week after the initial dosing, group 1 received an i.p. injection of 5% dextrose, and group 2 (LOHP alone) and group 3 (LOHP+EFVF) received LOHP (10 mg/kg) once a week, for 3 weeks. 

### 4.5. Assessment of Mechanical Allodynia 

To measure the pain sensitivity as a behavioral outcome, the response to external mechanical stimuli was determined using an electronic von Frey tester or plantar von Frey instrument. All tests were performed by an operator blinded to the information of the drug treatment in a quiet and controlled behavior test room between 2–5 PM. Animals were individually placed in a plastic chamber set on a perforated mesh-like open grid of square holes and acclimated for at least 30 min before the behavior test. 

In the treatment model, the nociceptive hypersensitivity to external stimuli was evaluated by measuring the stimulation force when the rat withdraws its hind paw using the Dynamic Plantar Aesthesiometer (DPA, Ugo Basile, VA, Italy). The von Frey-type 0.5 mm filament from a touch stimulator was applied on to the mid-plantar of the rat hind paw, and the stimulation force and speed were increased gradually (app. 5 g/s, cut-off force 50 g) and linearly until the rat withdrew its hind paw. The test was repeated 5 times with an interval of at least 5 min between each test cycle. The withdrawal threshold force (g) was calculated automatically by DPA.

In the prevention and treatment model, the response rate to external stimuli was measured using a calibrated von Frey monofilament with 0.16 and 0.4 g bending forces (DanMic Global, San Jose, CA, USA). The filament was applied five times to the mid-plantar surface of the mouse hind paw for 3 s with an interval of 10 s between stimulations. The test was performed to the hind paw with an interval of at least 15 min between each test cycle. The frequencies of paw withdrawal were calculated as the percentage of animals that showed a response (rapid and sudden lifting, shaking, or licking) occurring at least three times from a total of five trials.

### 4.6. Cell Culture and Cell Viability Assay

Human cancer cell lines, HCT116 and COLO205 for colon cancer, and MDA-MB-468 and MCF-7 for breast cancer were purchased the American Type Culture Collection (ATCC, Rockville, MD, USA). KM12SM colon cancer cells were obtained from the Korean Cell Line Bank (KCLB, Seoul, Republic of Korea). Cancer cells were cultured in the designated media supplemented with 10% (*v/v*) heat-inactivated fetal bovine serum (FBS) and an antibiotics mixture (penicillin/streptomycin) and maintained at 37 °C in a humidified CO_2_ incubator. The PC12 cell line, rat adrenal gland pheochromocytoma cells (ATCC), was cultured in a collagen type I-coated culture dish with DMEM supplemented with 10% heat-inactivated horse serum (HS) and 5% FBS and an antibiotics mixture. 

DRG cells were isolated as previously described [39]. Briefly, 6–8-week-old male mice (C57BL/6) were euthanized by CO_2_ inhalation. The skin was cut through the lumbar end of the spine. After removing the ribs, muscles and connective tissues, the spinal column was cut out and placed into L15 media. Under a dissecting stereo microscope (SZ-PT, Olympus, Tokyo, Japan), DRGs were plucked out and incubated in collagenase A and Dispase II (Roche, Mannheim, Germany) in 2 mL of L15 media in a water bath at 37 °C for 30 min. To isolate the cells, the DRGs were triturated and incubated with DNase I at 37°C for 30 min and then, purified with 30 and 60% gradient Percoll solutions in Hank’s balanced salt solution (HBSS). The DRG cells from the cloudy layer at the interface of the two Percoll solutions were cultured in Neurobasal/B27 on a poly-D-lysine/Laminin (PDL/L)-coated plate (Corning, Bedford, MA, USA). All culture media, sera, buffer solution, and other supplements for cell culture were obtained from Thermo Fisher Scientific (Grand Island, NY, USA).

For the cell viability assays, cells were seeded in a 96-well plate at a density of 5 × 10^3^ cells/well for the cancer cells or 2.5 × 10^3^ cells/well for the neurons, and allowed to attach overnight. The cells were exposed to the indicated concentrations of EFVF, LOHP (TOCRIS Bioscience, Bristol, UK), or a combination of LOHP with EFVF or amifostine (Santa Cruz Biotechnology, Dallas, TX, USA) for 48 h. An equal volume of PBS was added for the vehicle treatment. Cell proliferation was measured using the Ez-Cytox viability assay kit (Daeil Lab Service Co., Seoul, Republic of Korea). The relative proliferative index was calculated as the percentage of the experimental optical density (OD) value divided by the vehicle OD value. To determine the effect of EFVF on the values of the 50% cell growth inhibitory concentrations (IC_50_) of LOHP, cancer cells were co-treated with LOHP (0–100 μM) and EFVF (50 and 100 μg/mL) for 48 h, and IC_50_s were calculated using the SoftMax Pro 7.0 software (Molecular Devices, Sunnyvale, CA, USA).

### 4.7. Neurite Outgrowth Assay in PC12

Neurite outgrowth assay in PC12 was performed as described previously [39]. Briefly, PC12 cells (1 × 10^4^ cells/well) were cultured on a collagen type IV-coated 24-well plate for 24 h. To induce neurite growth, the cells were replaced with differentiation medium consisting serum-free DMEM, 100 ng/mL recombinant rat beta NGF (R&D Systems, Minneapolis, MN, USA), 1% N2 supplement (Thermo Fisher Scientific) and 0.5% FBS for 4 days. To investigate the effect of LOHP on neural differentiation of the PC12 cells, the cells were exposed to DM containing a combination of LOHP (200 nM), EFVF, and each constituent or amifostine (0.5 mM) as described earlier. A neurite outgrowth analysis was performed using a phase contrast bright-field inverted microscope (IX71, Olympus, Tokyo, Japan). Digitalized morphometric images of each well were obtained and the fields containing more than 20 cells were captured. The cells bearing neurite that had at least one neurite with a length longer than the diameter of the cell body were counted from four captured images and expressed as a percentage of the total of 80 cells counted from the images. Total neurite lengths were measured by manually tracing the length of the neurites using the Meta Morph image software (Molecular Devices).

### 4.8. Neurite Outgrowth Assay in DRGs

The neurite outgrowth assay in DRGs was conducted with the Neurite Outgrowth Assay Kit (NS225, Millipore, Billerica, MA, USA) according to the manufacturer’s instructions. Briefly, the membrane surface of Millicell inserts containing permeable membranes with 1 μm pores was pre-coated with PDL/L for 2 h at 37 °C and then, placed into culture plate wells containing 100 ng/mL NGF in the Neurobasal/B-27 medium. The DRG cells (2 × 10^4^) were added to the inserts and treated with LOHP and EFVF. After 7 days, the cells were rinsed and fixed with −20 °C methanol for 20 min at RT and then, the neurites were stained with the Neurite Stain Solution for 15–30 min at RT. After swabbing the cell bodies, the stained neurites were extracted on Parafilm using the Extraction Buffer for 5 min and quantified using a plate reader at 562 nm. 

### 4.9. Apoptosis Assay 

Apoptotic cells were simultaneously stained with fluorescein isothiocyanate (FITC)-labelled Annexin V (FITC-AV) and propidium iodide (PI) using the FITC Annexin V Apoptosis Kit (Life Technologies, Carlsbad, CA, USA). Mouse DRG cells (2 × 10^4^) were cultured in the Neurobasal/B27 medium with 100 ng/mL NGF on a sterile 12 mm round coverslip pre-coated with PDL/L. After a week, the cells were exposed to LOHP (30 μM) with or without EFVF for 72 h and then, washed twice with cold PBS. The cells were incubated with Annexin V/PI working solution in Annexin-Binding Buffer (ABB) for 15 min at RT in the dark place and then, washed with ABB. The cells were observed under an inverted microscope using a fluorescence system (U-RFL-T, Olympus) with FITC/tetramethylrhodamine (TRITC) filters. The cells undergoing late apoptosis show red and green fluorescence, and live healthy cells show no fluorescence. 

### 4.10. Mitochondrial Membrane Potential (MMP) Assay

Changes in the MMP were determined using a cyanine dye JC-1 (5,5′,6,6′-tetrachloro-1,1′,3,3′-tetraethyl-benimidazolocarbocyanine iodide, Molecular Probes, Eugene, OR, USA), a mitochondrial potential sensor that exhibits potential- dependent accumulation in mitochondria [56]. Briefly, mouse DRG cells (2 × 10^4^) were cultured in the medium containing 100 ng/mL NGF on a sterile 12 mm coverslip pre-coated with PDL/L for 1 week. Cells were treated with LOHP (30 μM) in the absence or presence of EFVF (25 and 50 μg/mL) for 24 h and then, cells were washed with PBS and stained with 2 μg/mL JC-1 for 20 min at 37 °C in the dark. The presence of JC-1 monomers or dimer aggregates was examined under a fluorescent microscope with standard FITC/TRITC filter sets of 530 nm/590 nm (aggregate) and 485 nm/538 nm (monomers). The conversion of red to green fluorescence of JC-1 shows the collapse in MMP.

### 4.11. Measurement of ROS Formation

The detection of ROS production was performed using the CellROX Green Reagent (Life Technologies) according to the manufacturer’s instructions. Briefly, neural PC12 cells were cultured with drugs for 24 h, washed with PBS and then, stained with the 5 µM CellROX Green Reagent at 37 °C for 1 h. The fluorescence intensity was measured with a fluorescence microplate reader at an excitation wavelength of 485 nm and emission wavelength of 535 nm. 

### 4.12. Immunohistochemistry 

After the final behavior test, the rats were deeply anesthetized with a pentobarbital solution (50 mg/kg) and subjected to transcardial perfusion of 4% paraformaldehyde (PFA) using a peristaltic pump (120S, Watson Marlow, Cornwall, UK). The hind footpads were removed from the rat and fixed in 4% PFA. The paraffin-embedded hind footpads were sectioned at a 4 μm thickness and then, mounted on silane-coated slides (Muto Pure Chemicals, Tokyo, Japan). After deparaffinization, the sections were washed in Tris-buffered saline with 0.5% tween 20 (TBS-T) and then, incubated with a primary antibody against PGP9.5, as a marker of IENF (1:200, Millipore, Temecula, CA, USA) at 4°C overnight. After washing with TBS-T, the sections were incubated with the secondary anti-rabbit antibody labeled with Alexa Flour 488 (Abcam) for 1 h at RT and covered with a mounting medium (Vector shield, Vector laboratory, Burlingame, CA, USA). The mounted slide was photographed under a fluorescence microscope (200× magnification), and the images were analyzed with image software (Cellsense, Olympus). Quantification of the IENF density was done by counting IENFs crossing the dermal-epidermal junction from 3 randomly chosen slices per foot pad. The number of IENFs per sight was counted in 2 fields of view from each slice. The relative IENF density index was calculated as the percentage of IENF number divided by that of the vehicle treatment.

### 4.13. Statistical Analysis 

Each in vitro experiment was performed at least three times, and the values are expressed as the means ± standard deviations (SD). One-way ANOVA and two-way repeated measures ANOVA were performed for comparing the means followed by a Tukey’s post hoc multiple comparison. All statistical analyses were performed with the SigmaPlot 13.0 software (Systat Software, San Jose, CA USA). Significant differences were considered at *p* < 0.05. 

## Figures and Tables

**Figure 1 molecules-24-01177-f001:**
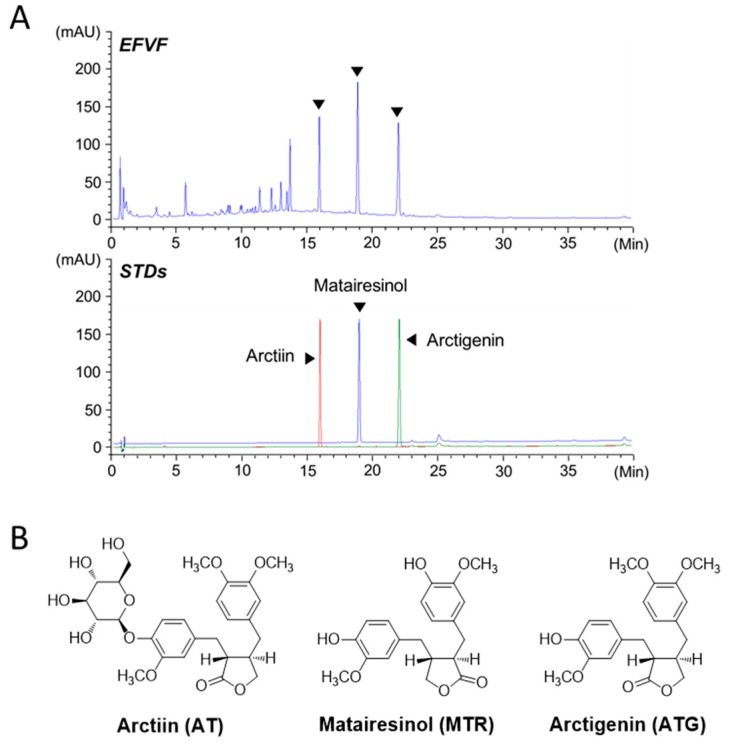
UHPLC chromatograms of EFVF and its three major constituents. (**A**) The DAD signals were detected at 280 nm. (**B**) The structure of the three authentic standard chemicals: AT, MTR, and ATG.

**Figure 2 molecules-24-01177-f002:**
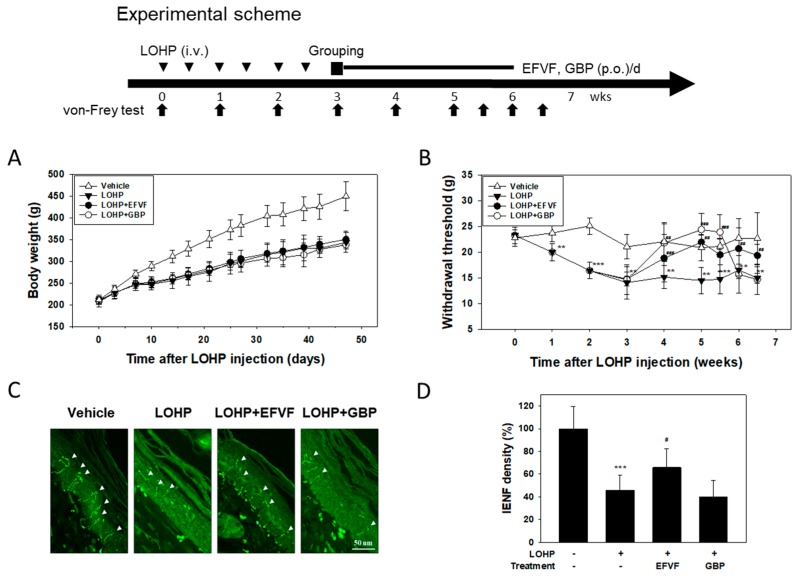
Neuroprotective effects of EFVF against LOHP-induced mechanical hyperalgesia. Rats in each group (*n* = 6) received LOHP (5 mg/kg, i.v., 6 times), EFVF (100 mg/kg, p.o.) and GBP (50 mg/kg, p.o.) as shown in the experimental scheme. 5% (*w/v*) dextrose was given to the vehicle control group. (**A**) Body weights were measured every three or four days throughout the experiment. (**B**) Withdrawal threshold was assessed by the electrical von Frey test. Each point represents the mean ± SD of six rats. * *p* < 0.05, ** *p* < 0.01, *** *p* < 0.001 vs. vehicle control and ^#^
*p* < 0.05, ^##^
*p* < 0.01, ^###^
*p* < 0.001 vs. LOHP alone. Significant difference among groups at each indicated time point (two-way repeated measures ANOVA with Tukey’s post hoc multiple comparison; in Group, DF = 3, *F* = 10.914, *p* < 0.001; in Time DF = 8, *F* = 9.850, *p* < 0.001; Group × Time DF = 24, *F* = 3.556, *p* < 0.001). (**C**) At the end of the experiment, the footpads of the rats were procured and subjected to immunohistochemical analyses with a pan-axonal marker PGP9.5. The immunoreactive IENFs are indicated by arrow heads (×400 magnification). (**D**) Innervation of the IENFs through the rat footpad skin were quantified by counting IENFs under fluorescence microscopy using FITC filters dividing the number of IENFs by the length of the section and then, expressed as relative densities of IENFs compared to vehicle. *** *p* < 0.001 vs. vehicle control group. ^#^
*p* < 0.05, vs. LOHP alone.

**Figure 3 molecules-24-01177-f003:**
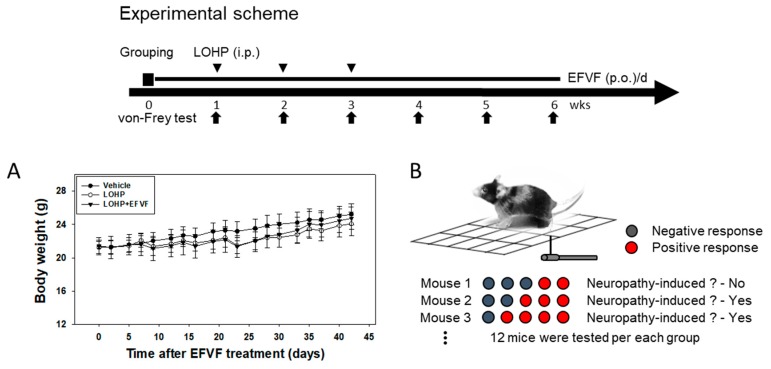
Preventive effect of EFVF against mechanical hyperalgesia induced by LOHP. As depicted in the experimental schedule, EFVF (50 mg/kg, p.o.) was daily administered for 6 weeks. From one week after the first EFVF treatment, LOHP (10 mg/kg, IP, 3 times) was administered once a week, for three weeks. As an untreated control, the vehicle groups were treated with 5% dextrose. (**A**) Body weights were measured every two or three days. (**B**) At the indicated time point (arrow), a mechanical stimulus was applied five times on the plantar of the right and left hind paws using a von Frey monofilament with a 0.16 g or 0.4 g bending force, respectively. The incidence rates of each group (*n* = 12) were scored by counting the number of animals that responded more than three times to five applications with 10 s intervals. The results are presented in Table 2.

**Figure 4 molecules-24-01177-f004:**
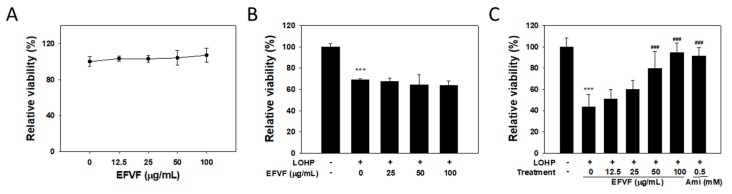
EFVF reduced the LOHP-induced neurotoxicity in neural PC12 cells. (**A**) Cells were cultured with various concentrations of EFVF for 48 h in the absence of NGF. (**B**) Cells were cultured with NGF (100 ng/mL), LOHP (200 nM), and EFVF for 48 h. (**C**) Cells were pre-cultured in differentiation medium containing 100 ng/mL NGF for 96 h and then, treated with a combination of LOHP (200 nM) and EFVF (0–100 μg/mL) for a further 48 h. Amifostine (Ami, 0.5 mM) was used as a positive control. (**A**–**C**) Cell viability was measured with the Ez-Cytox viability assay kit. The data from triplicate experiments are expressed as the mean ± SD. *** *p* < 0.001 vs. vehicle control (-/-), and ^###^
*p* < 0.001 vs. LOHP alone.

**Figure 5 molecules-24-01177-f005:**
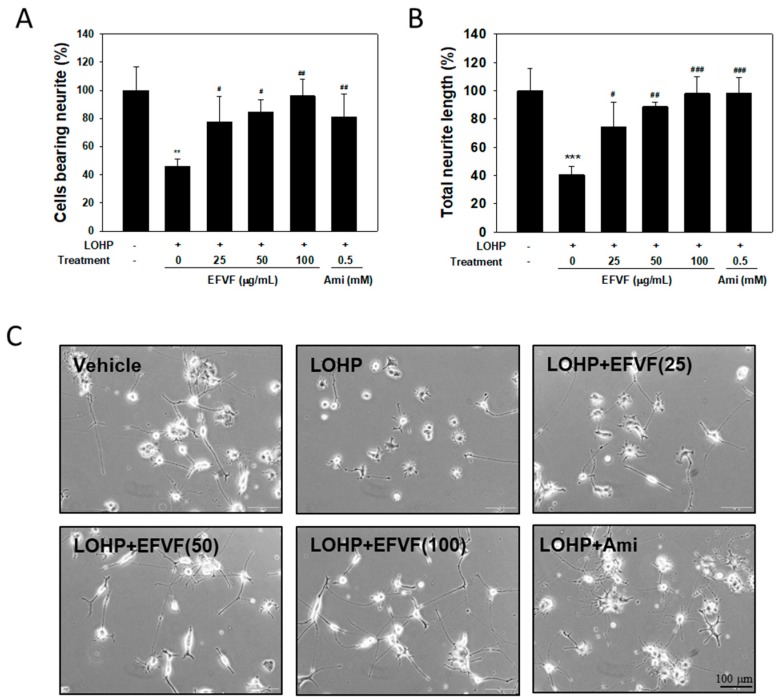
EFVF attenuated the LOHP-mediated neurotoxicity during the differentiation of PC12 cells. Cells were cultured with NGF (100 ng/mL) in a type IV collagen-coated 24-well plate in the presence or absence of LOHP (200 nM) and EFVF (0–100 μg/mL) for 72 h. Amifostine (Ami, 0.5 mM) was used as a positive control. On day 3 of the culture, the neurites sprouting from the cell bodies were counted under an inverted microscope. The percentages of cells bearing neurites (**A**) and their total neurite lengths (**B**) were quantified using an image analyzing software. Data from triplicate experiments represent the mean ± SD. (**C**) Images were photographed under an inverted microscope (×200 magnification). ** *p* < 0.01, *** *p* < 0.001 vs. vehicle control group. ^#^
*p* < 0.05, ^##^
*p* < 0.01, ^###^
*p* < 0.001 vs. LOHP alone.

**Figure 6 molecules-24-01177-f006:**
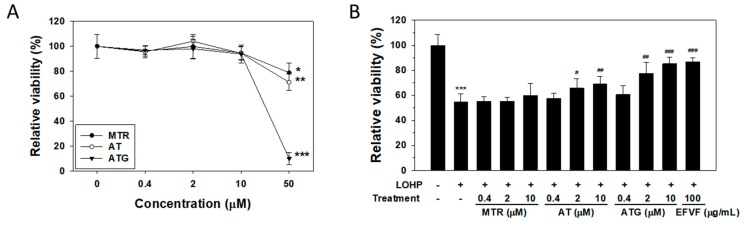
Effect of the three constituents of EFVF on the LOHP-induced neurotoxicity in neural PC12 cells. (**A**) Cells were cultured with various concentrations of MTR, AT, and ATG, 48 h. (**B**) Cells were cultured in differentiation medium containing NGF (100 ng/mL) for 96 h and then, treated with LOHP (200 nM) alone or in combination with the indicated amounts of compounds for a further 48 h. EFVF (100 μg/mL) was used as a positive control. Cell viability was measured with the Ez-Cytox viability assay. The data from triplicate experiments are expressed as the mean ± SD. * *p* < 0.05, ** *p* < 0.01, *** *p* < 0.001 vs. vehicle control, and ^#^
*p* < 0.05, ^##^
*p* < 0.01, ^###^
*p* < 0.001 vs. LOHP alone.

**Figure 7 molecules-24-01177-f007:**
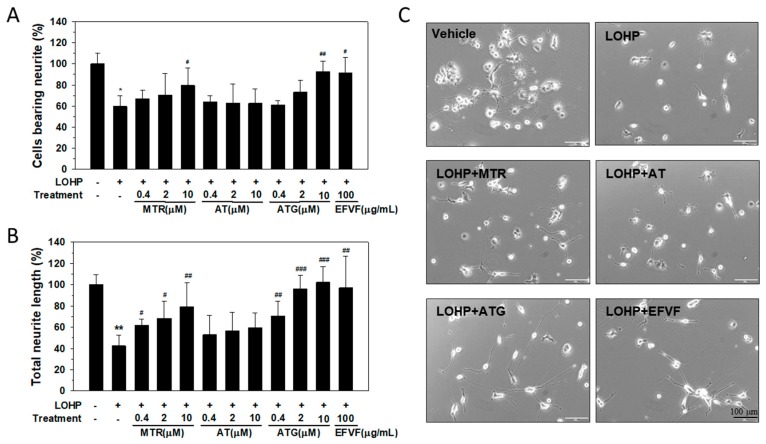
Effect of the three constituents of EFVF on the LOHP-induced neurotoxicity of neural PC12 cells. The cells were cultured in differentiation medium containing NGF (100 ng/mL) and treated with LOHP (200 nM) alone or in combination with the indicated concentration of MTR, AT, and ATG for 72 h. EFVF (100 μg/mL) was used as a positive control. On day 3, the extended neurites were counted under an inverted microscope. The percentage of cells bearing neurites (**A**) and their total neurite lengths (**B**) were quantified using an image analyzing software. Data from triplicate experiments represent the mean ± SD. (**C**) Images were photographed under an inverted microscope (×200 magnification). * *p* < 0.05, ** *p* < 0.01, *** *p* < 0.001 vs. vehicle control ^#^
*p* < 0.05, ^##^
*p* < 0.01, ^###^
*p* < 0.001 vs. LOHP alone.

**Figure 8 molecules-24-01177-f008:**
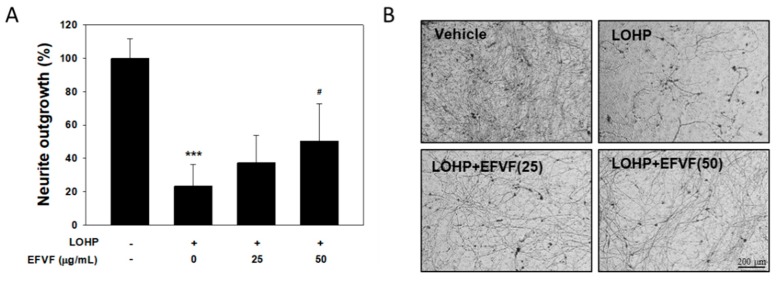
Neuroprotective effect of EFVF against LOHP-induced neurotoxicity in primary DRG cells. (**A**) DRG cells were suspended in differentiation medium containing NGF and then, the cells (2 × 10^4^) were plated on the wells of a Neurite Outgrowth Plate and treated with LOHP or EFVF. After 7 days, the neurites were stained with Neurite Stain Solution and extracted using Extraction Buffer. Neurite outgrowth was quantitated using a microplate reader at 562 nm. The results are presented as a relative percentages compared to a saline-treated vehicle group. *** *p* < 0.001 vs. vehicle control, ^#^
*p* < 0.05 vs. LOHP alone. (**B**) The representative images of neurites were photographed under an inverted microscope (×100 magnification).

**Figure 9 molecules-24-01177-f009:**
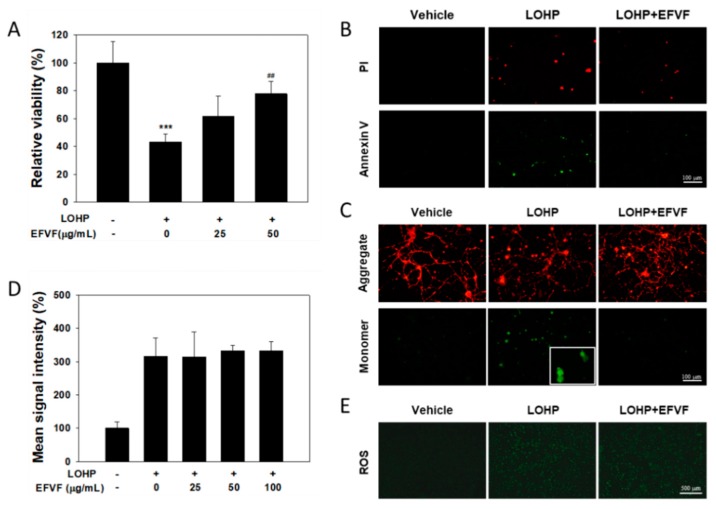
Effects of EFVF on neural cell viability, mitochondria dysfunction, and ROS production. (**A**) DRG cells (2.5 × 10^3^) were incubated with NGF (100 ng/mL) in the indicated concentrations of EFVF for 48 h. Cell viability was measured with the Ez-Cytox viability assay. (**B**,**C**) DRG cells (2 × 10^4^) were cultured on PDL/L-coated sterile 12 mm coverslips under the same condition as described in (**A**). (**B**) To analyze apoptosis, the cells were stained with Annexin V and PI solution for 15 min at RT in the dark. Stained cells were observed under a fluorescence microscopy with FITC/TRITC filters (×200 magnification). (**C**) To analyze mitochondria dysfunction, cells were stained with 2 μg/mL JC-1 for 20 min at 37 °C in the dark. The presence of JC-1 monomers or dimers was examined by fluorescent microscope (×200, and ×400 magnification for insert). (**D**,**E**) PC12 cells were culture with LOHP alone or in combination with EFVF for 24 h and then, stained with CellROX Green reagent for 1 h. The mean signal intensity was measured with a fluorescence multi-plate reader at 485/520 nm excitation and emission (**D**). The cells producing ROS showed green fluorescence under the fluorescence microscope (×40 magnification, E). Data are presented as the mean ± SD from three experiments. *** *p* < 0.001 vs. vehicle control, ^##^
*p* < 0.01 vs. LOHP alone.

**Table 1 molecules-24-01177-t001:** Summary of quantitation parameters.

Constituents	Linear Range (μg/mL)	Regression Equation	r^2^	LOD (μg/mL)	LOQ (μg/mL)
Arctiin	5–200	y=6110.2x+3.3480	0.9999	0.33	0.99
Matairesinol	5–200	y=4308.9x+1.5386	0.9999	0.12	0.37
Arctigenin	5–200	y=8834.4x−0.1731	0.9998	0.21	0.62

LOD, limit of detection; LOQ, limit of quantitation.

**Table 2 molecules-24-01177-t002:** The incidence rates of the mechanical hypersensitivity.

Treatment	N	Hind Paw (N)	Time (Week)					
			1	2	3	4	5	6
**0.16 g Bending Force Filament**								
Vehicle	12	Left	2	3	2	2	2	2
		Right	1	1	3	3	3	2
		Avg (%) ^1^	12.5	16.7	20.8	20.8	20.8	16.7
LOHP	12	Left	2	4	6	6	5	6
		Right	1	2	6	6	5	4
		Avg (%) ^1^	12.5	25.0	50.0	50.0	41.7	41.7
LOHP+EFVF	12	Left	1	3	3	4	1	2
		Right	2	3	3	3	4	4
		Avg (%) ^1^	12.5	25.0	25.0	29.2	20.8	25.0
**0.4 g Bending Force Filament**								
Vehicle	12	Left	3	4	5	4	4	4
		Right	4	5	3	4	5	4
		Avg (%) ^1^	29.2	37.5	33.3	33.3	37.5	33.3
LOHP	12	Left	4	8	9	9	9	7
		Right	2	6	9	10	9	7
		Avg (%) ^1^	25.0	58.3	75.0	79.2	75.0	58.3
LOHP+EFVF	12	Left	3	4	3	4	7	4
		Right	3	5	5	4	3	6
		Avg (%) ^1^	25.0	37.5	33.3	33.3	41.7	41.7

^1^ Average values were expressed as the mean of the incidence rates in the left and right footpads.

**Table 3 molecules-24-01177-t003:** Effect of EFVF on the cytotoxicity of LOHP in human cancer cell lines.

Treatment	IC_50_ Value (μM) ^1^				
	HCT116	COLO205	KM12SM	MDA-MB-468	MCF7
Vehicle	1.6 ± 0.1	2.0 ± 1.4	4.9 ± 1.0	4.9 ± 0.2	7.2 ± 3.5
EFVF 50 μg/mL	2.1 ± 0.9	2.2 ± 1.8	5.9 ± 2.0	4.8 ± 0.4	6.5 ± 3.2
EFVF 100 μg/mL	2.8 ± 1.2	1.9 ± 1.5	4.4 ± 0.5	4.6 ± 0.1	6.8 ± 0.3

^1^ IC_50_ value is presented as the mean ± SD from triplicate experiments.

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
