# Peer review of "Neuroprotective Effects of an Aqueous Extract of Forsythia viridissima and Its Major Constituents on Oxaliplatin-Induced Peripheral Neuropathy"

_molecules, 2019, doi:10.3390/molecules24061177_

Round 1

Reviewer 1 Report

The authors show in this manuscript a study of the effect of Oxaliplatin (LOHP) in presence of an aqueous Extract of Forsythia Viridissima. This is an interesting paper where it is supplied some data of interest.

Consequently, this work should be suitable for publication in Molecules. However, major changes have to take into account:

The authors show in results that to determine the major constituents of EFVF, its UHPLC chromatogram was scanned and compared to the peaks of authentic standard chemicals (STDs). How were these STDs chosen?. The pharmacological values and chemical compositions of F. viridissima have not been elucidated fully, according to the “Introduction”. I do not understand how the authors knew that the appropriate standard chemicals had to be arctiin (AT), matairesinol (MTR), and arctigenin (ATG), and why they did not choose other chemicals. In addition, it is impossible to identify the unknown chemical constituents of EFVF with a diode array detector because it is impossible to choose the appropriate standard chemicals, it is needed, at least, a mass spectrometer coupled to the UHPLC and an appropriate database performed by the users. All these uncertainties have to be clarified.

In animal assays, how did the authors choose the dose?  I mean, why did Sprague-Dawley rats receive 5 mg/kg of LOHP or 100 mg/kg of EFVF?, why did mice receive 10 mg/kg of LOHP or 50 mg/kg of EFVF?, what was the criterion to use these doses?. Furthermore, the EFVF dose is missed in the text in mice assay.

Minor changes

In page 6 lines 177-178, please review the following sentence ”PC12 cells cultured….. of EFVF for 48 h”, it makes no sense.

In page 6 line 202, “that” is duplicated

Author Response

Response to Reviewer 1 Comments

The authors show in this manuscript a study of the effect of Oxaliplatin (LOHP) in presence of an aqueous Extract of Forsythia Viridissima. This is an interesting paper where it is supplied some data of interest.

Consequently, this work should be suitable for publication in Molecules. However, major changes have to take into account:

Comment #1.

The authors show in results that to determine the major constituents of EFVF, its UHPLC chromatogram was scanned and compared to the peaks of authentic standard chemicals (STDs). How were these STDs chosen? The pharmacological values and chemical compositions of F. viridissima have not been elucidated fully, according to the “Introduction”. I do not understand how the authors knew that the appropriate standard chemicals had to be arctiin (AT), matairesinol (MTR), and arctigenin (ATG), and why they did not choose other chemicals. In addition, it is impossible to identify the unknown chemical constituents of EFVF with a diode array detector because it is impossible to choose the appropriate standard chemicals, it is needed, at least, a mass spectrometer coupled to the UHPLC and an appropriate database performed by the users. All these uncertainties have to be clarified.

Response # 1

Although the pharmacological values and chemical compositions of F. viridissima have not been fully elucidated, a range of phytochemicals including phenylethanoid glycosides, lignans, iridoids, flavonoids, and triterpenoids, have been reported from the methanol extracts of Forsythia viridissima flowers, roots or fruits. Among them, the biological activities such as anti-inflammation, antihistamine, and antivirus by lignans have been studied well (1-3).

Huh, J.; Song, J. H.; Kim, S. R.; Cho, H. M.; Ko, H. J.; Yang, H.; Sung, S. H., Lignan Dimers from Forsythia viridissima Roots and Their Antiviral Effects. J Nat Prod 2019, 82(2), 232-238.

Lee, J.H. Lee JY., Kim T.D. and Kim C.J. Antiasthmatic Action of Dibenzylbutyrolactone Lignans from Fruits of Forsythia viridissima on Asthmatic Responses to Ovalbumin Challenge in Conscious Guinea-pigs. Phytother. Res. 2001, 25, 387–395.

Tokar, M.; Klimek, B., Isolation and identification of biologically active compounds from Forsythia viridissima flowers. Acta poloniae pharmaceutica 2004, 61(3), 191-197.

Therefore, to determine the major constituents of EFVF, we first obtained the UHPLC chromatogram of EFVF and then set up the phytochemical DAD database which contained information of the DAD spectra and retention times (tR) of lignan family chemicals including arctiin, arctigenin, matairesinol, phillygenin, phillyrin, pinoresinol, (+) pinoresinol-β-D-glucopyranoside, caffeoyl pinoresinol, lariciresinol, using standard chemicals.

Secondly, the UHPLC chromatogram of EFVF was scanned and compared to the DAD spectra of lignan family. We found that the three major peaks of EFVF showed typical DAD spectra similar to those of arctiin (AT), matairesinol (MTR), and arctigenin (ATG).

Finally, we confirmed that the tR of authentic standard chemicals (STD) of AT, MTR, and ATG matched those of the three major constituent peaks of EFVF. In addition, the DAD spectra in 210-450 nm spectral scanning also showed a perfect matching between the major constituents of EFVF and their STDs (supplement Figure 1).

We described the UHPLC analysis of EFVF in more detail in the Results section, 2.1.  (the revised manuscript, Page 2, lines 79-89) and added Supplement Figure 1 showing DAD spectra of EFVF major peaks and their STDs in 210 and 450 nm spectral scanning.

“Since several lignans from the methanol extract of F. viridissima were isolated and their biological activities have been studied, we first compared the DAD spectra and retention times (tR) data between EFVF peaks and standard chemicals of lignan family, including arctiin, arctigenin, matairesinol, phillygenin, phillyrin, pinoresinol, (+) pinoresinol-β-D-glucopyranoside, caffeoyl pinoresinol, lariciresinol. Among them, the three major peaks of EFVF showed typical DAD spectra similar to those of lignans such as arctiin (AT), matairesinol (MTR), and arctigenin (ATG). The retention times (tR) of AT, MTR, and ATG in the overlaid chromatogram of STDs (Figure 1A, lower panel) were 15.91, 18.90, and 21.98 min, respectively, and they matched those of the three major constituent peaks of EFVF. In addition, the DAD spectra in 210 and 450 nm spectral scanning also showed a perfect matching between the three constituents of EFVF and their STDs (supplement Figure 1).”

Comment #2. In animal assays, how did the authors choose the dose?  I mean, why did Sprague-Dawley rats receive 5 mg/kg of LOHP or 100 mg/kg of EFVF? Why did mice receive 10 mg/kg of LOHP or 50 mg/kg of EFVF? What was the criterion to use these doses?. Furthermore, the EFVF dose is missed in the text in mice assay.

Response #2

Mechanical hypersensitivity has been observed in the murine animal models by  intraperitoneal and intravenous injection of oxaliplatin of about 30 mg/kg accumulative doses (1-3). For example, oxaliplatin (4 mg/kg) was administered intravenously twice per week for 4 weeks (total 32 mg/kg), and mechanical allodynia was evaluated using the von Frey test in rats (1).

Similarly, in our present study, rat and mice received total accumulated dose of 30 mg/kg oxaliplatin (rat, 5mg/kg, twice per week for 3 weeks; mice, 10 mg/kg, once per week for 3 week) for induction of chronic OIPN model as explained in the “Materials and method” (page 15, lines 535 – 539 and lines 540-545) and “Figure Legend”  (page 4, line 139 and page 6, line 174 in the revised manuscript ).

Fujita, S., Ushio, S., Ozawa, N., Masuguchi, K., Kawashiri, T., Oishi, R., and Egashira, N. Exenatide Facilitates Recovery from Oxaliplatin-Induced Peripheral Neuropathy in Rats. PLoS One. 2015, 10(11), e0141921-(ref 31, in the revised Manuscript)

Hache G, Guiard BP, Nguyen TH, Quesseveur G, Gardier AM, Peters D, Munro G, Coudoré F. Antinociceptive activity of the new triple reuptake inhibitor NS18283 in a mouse model of chemotherapy-induced neuropathic pain. Eur J Pain. 2015,19(3):322-33.

Renn, C. L.; Carozzi, V. A.; Rhee, P.; Gallop, D.; Dorsey, S. G.; Cavaletti, G., Multimodal assessment of painful peripheral neuropathy induced by chronic oxaliplatin-based chemotherapy in mice. Mol pain 2011, 7, 29. (ref 43, in the revised Manuscript)

In Korean traditional herbal medicine, Forsythiae Fructus (FF), the dried fruits of F. suspensa and F. viridissima has been generally used 3 ~15 g as a daily dosage (1). Since the yield of EFVF was 16.8% (w/w) in the present study, based on the average person who weighs 60 kg, it actually taken about 8.4-41 mg/kg. Therefore, we calculated the initial dosage for murine study using recommended doses of FF to evaluate its relief effect against oxaliplatin-induced peripheral neuropathy. We chose above 50 ~ 100 mg/kg of EFVF for murine as an initial dose which is equivalent to 8.4 mg/kg in an adult human (2).

Shin MK. Clinical Traditional Herbalogy. 5th ed.; YoungLim’s publisher: Seoul, Republic of Korea; 1996; pp. 322.

US FDA Guidance for Industry; Estimating the maximum safe starting dose in initial clinical trials for therapeutics in adult healthy volunteers, 2005. Pharmacology and Toxicology. http://www.fda.gov/cder/guidance/ index.htm

To evaluate the treatment effect of EFVF on peripheral neuropathy induced by oxalipaltin, we first administered EFVF (100 & 300 mg/kg) daily for 3 weeks to rats after oxalipaltin treatment. The withdrawal threshold of rats was dramatically decreased to 14.5 ± 0.4 g by oxalipaltin treatment, which were increased to 18.8 ± 4.7 g by treatment of EFVF (100 mg/kg, Figure 2B). Additionally, the treatment of EFVF (300 mg/kg) also attenuated oxalipaltin-induced decrease in the withdrawal threshold to the similar level of those of 100 mg/kg EFVF group, and then decreased again to the basal level, which was shown in GBP treated group (as shown in Figure A below).

So, we presented the treatment efficacy of EFVF at dose 100 mg/kg (rat) as shown in Figure 2.

Next we evaluated the preventive effect of EFVF against mechanical hyperalgesia induced by oxaliplatin by co-treatment of oxaliplatin with EFVF (50 and 200 mg/kg, p.o.) daily for 6 weeks to mice. Interestingly, we found that co-treatment of oxaliplatin with EFVF (50 mg/kg) reduced the mechanical hypersensitivity induced by oxaliplatin throughout the experiment: the response rate was decreased again to basal level compared to the vehicle mouse (Figure 3B & Table 2). However, co-treatment of oxaliplatin with EFVF (200 mg/kg) showed a little change in the mechanical hypersensitivity induced by oxaliplatin (as shown in Figure B, below).

So, we presented the preventive efficacy of EFVF at dose 50 mg/kg (mouse) as shown in Figure 3 and Table 2.

Figure A                                 Figure B

We described in Materials and Method section, 4.4 as follows and added new references (54 & 55); In Korean traditional herbal medicine, Forsythiae Fructus of 3~15 g has been generally recommended as a daily dosage for heat clearing and detoxifying medication. Based on the average person who weighs 60 kg, it was actually taken about 8.4 ~ 41 mg/kg of EFVF (16.8 % yield). Therefore, we chose about 50 ~ 100 mg/kg as a starting dose for murine studies, which is equivalent to about 8.4 mg/kg in an adult human (Page 15, lines 530-534, in the revised manuscript).

According to the reviewer’s comment, we added the EFVF dose (50 mg/kg) for administration of mice into page 5, line 156 in the revised manuscript.

Comment #3.  Minor change

In page 6 lines 177-178, please review the following sentence ”PC12 cells cultured….. of EFVF for 48 h”, it makes no sense.  In page 6 line 202, “that” is duplicated

Response #3

As indicated by the reviewer, the sentence PC12 cells were cultured in a differentiation medium containing NGF with oxaliplatin (200 nM), and the indicated concentration of EFVF for 48 h was corrected as follows. PC12 cells were cultured in a differentiation medium containing NGF, and treated with a combination of LOHP (200 nM) and indicated concentration of EFVF for 48 h (Page 6, lines 191-192, in the revised manuscript).

As indicated by the reviewer, in page 6 line 216, duplicated “that” is removed. (Page 7, line 216, in the revised manuscript)

Reviewer 2 Report

This paper investigated the effects of Forsythia Viridissima on oxaliplatin-induced peripheral neuropathy. The introduction and discussion were well narrated, and the experiments were properly designed. However, several defects must be corrected for acceptance.

1. In Fig. 2B, this figure was acquired and measured from the same rats during different intervals; therefore, the analysis using two-way ANOVA with repeated measurement was required. Besides, if proper, the degree of freedom and F value should be added after each analysis.

2. In Fig. 3, no statistical analysis was performed and described?

Author Response

Response to Reviewer 2 Comments

This paper investigated the effects of Forsythia Viridissima on oxaliplatin-induced peripheral neuropathy. The introduction and discussion were well narrated, and the experiments were properly designed. However, several defects must be corrected for acceptance.

Comment #1.  In Fig. 2B, this figure was acquired and measured from the same rats during different intervals; therefore, the analysis using two-way ANOVA with repeated measurement was required. Besides, if proper, the degree of freedom and F value should be added after each analysis.

Response #1 As reviewer’s comment, the difference among groups at the indicated time points was analyzed using two-way repeated measures ANOVA followed by a Tukey’s post hoc multiple comparison. The degrees of freedom and F values after each analysis were also included in the Result section 2.2 and Figure 2(B) legend as follows;

Result section, 2.2: In addition, there was a significant interaction between treatment group and time (two-way repeated measures ANOVA, p = <0.001). (page 4, lines 125-126, in the revised manuscript)

Figure 2B, legend: Significant difference among groups at each indicated time point (two-way repeated measures ANOVA with a Tukey’s post hoc multiple comparison; in Group, DF= 3, F = 10.914, p < 0.001; in Time DF= 8, F = 9.850, P < 0.001; Group x Time DF= 24, F = 3.556, P < 0.001). (Page 4, line 144 – page 5, line 146 in the revised manuscript).

Comment #2. In Fig. 3, no statistical analysis was performed and described?

Response #2

To observe the preventive effect of EFVF against mechanical hyperalgesia induced by oxaliplatin, the number of animals that showed a paw withdrawal in response to mechanical stimuli were counted in each groups treated with vehicle, oxaliplatin, and oxaliplatin + EFVF as shown in the new Figure 3(B), and then the incidence rates of each group (n = 12) were scored by counting the number of animals that responded more than three times to 5 times applications. The results were presented in the new Table 2 (Page 6, lines 182-183 in the revised manuscript).

We corrected the Results Section 2.3 as follows; The pain sensitivity was measured by the von Frey test as shown in Figure 3B. The incidence rates of mechanical hypersensitivity in each treatment group were calculated by counting the number of mice that withdrew the left and right hind paw in response to mechanical stimuli. LOHP, 2-3 weeks after treatment dramatically increased the incidence rates of hypersensitivity to mechanical stimuli by von Frey filaments with 0.16 and 0.4 g bending forces up to 50.0 and 75.0 - 79.2%, respectively, while the untreated vehicle mice showed response rates below 20.8 and 37.5%, respectively. Interestingly, co-treatment of EFVF (50 mg/kg) reduced the mechanical hypersensitivity induced by LOHP throughout the experiment: the incidence rates were decreased again to basal level compared to the vehicle mouse (Table 2). (Page 5, lines 158 – 166, in the revised manuscript)

Round 2

Reviewer 1 Report

Nothing to comment.